# Advanced Technologies for Offering Situational Intelligence in Flood Warning and Response Systems: A Literature Review

**Srimal Samansiri** [1,*]**, Terrence Fernando** [1] **and Bingunath Ingirige** [2]

1 Think Lab, School of Science, Engineering and Environment, University of Salford, Salford M5 4WT, UK; t.fernando@salford.ac.uk

2 Centre for Disaster Resilience, School of Science, Engineering and Environment, University of Salford, Salford M5 4WT, UK; m.j.b.ingirige@salford.ac.uk

* Correspondence: r.s.p.samansiri@edu.salford.ac.uk

**Abstract:** Deaths and property damage from floods have increased drastically in the past two decades due to various reasons such as increased populations, unplanned developments, and climate change. Such losses from floods can be reduced by issuing timely early warnings and through effective response mechanisms based on situational intelligence during emerging flood situations. This paper presents the outcome of a literature review that was conducted to identify the types and sources of the intelligence required for flood warning and response processes as well as the technology solutions that can be used for offering such intelligence. Twenty-seven different types of intelligence are presented together with the technologies that can be used to extract such intelligence. Furthermore, a conceptual architecture that illustrates how relevant technology solutions can be used to extract intelligence at various stages of a flood cycle for decision-making in issuing early warnings and planning responses is presented.

**Keywords:** flood early warning and response processes; situational intelligence; conceptual architecture





## 1. Introduction

Floods are a frequently occurring hazard that impose adverse effects on a significant number of humans and cause substantial economic damage worldwide [1]. They constitute the highest number of recorded hazards and are responsible for 43% of total disasters between 1998 and 2017 according to the joint report by the Centre for Research on the Epidemiology of Disasters (CRED) and the United Nations Office for Disaster Risk Reduction [2]. Floods affected 2 billion people and caused 142,000 deaths during this period [2] and in recent years, flood frequency and its impact have been increasing drastically due to climate change and unplanned urban developments [3,4].

Panwar and Sen [5] suggest that the economic impact of natural hazards such as floods is more prominent in developing countries, and the CRED/UNISDR report [2,6] discloses that deaths from natural hazards in low-income countries are seven times higher than those in high-income countries. The key contributing factors for such increased losses have been recognised as population growth and rapid urbanisation [7,8]. Many researchers [6,9] have asserted that such losses and causalities can be significantly reduced by implementing an effective Flood Early Warning and Response Systems (FEWRS). In this regard, the Sendai Framework for Disaster Risk Reduction (SFDRR) emphasises the need for the availability of multi-hazard warning systems and disaster risk information to the community by the end of 2030 [10] (p. 12). The SFDRR promotes the need to have an integrated and coordinated approach to "generate, process and disseminate" disaster risk information using state-of-art technologies as a priority in member countries [11].

A study conducted by Rogers [12] reports that an effective forecast and warning system based on accurate real-time information on disasters can reduce the average annual flood damage by up to 35%. Furthermore, Seng [13] asserts that such a system can reduce

vulnerability and mortality rates. The reasons for the continued use of ineffective early warning systems that cause higher death rates are considered to be due to bureaucratic water management and digital divide-related issues [14], resulting in a lack of timely information for issuing warnings [9]. Therefore, the availability of an information system (IS) that can offer accurate and timely data with high service quality and user satisfaction has been recognised as one of the key reasons for implementing efficient FEWRSs [9]. Such information systems for flood warning and response systems should offer hazard detection, forecasting, warnings, and responses [15]. A study of global early warning systems within developed and developing countries [16,17] has found that the availability of such technology platforms is an important factor that influences the effectiveness of early warning systems in addition to policy, institutional, and societal factors.

Situational intelligence is crucial for making sound decisions quickly and efficiently, especially in a crisis situation [18]. In simple terms, situational intelligence is the ability to anticipate and react in a given situation [19]. Schilling argues that situational intelligence is a process that offers insight into a situation by processing and visualising multi-sourced data to make an appropriate decision on a specific situation [20]. The concept of situational intelligence has been used in numerous applications including the military [21,22], power and energy [23,24], petroleum [25], aviation [26], and disaster response [27,28]. In a crisis scenario, authorities need credible information to understand the reality on the ground. For example, in fluvial flooding, rainfall, river water level, and the number of persons impacted are typical types of intelligence that are required by authorities for making better decisions when issuing early warnings and responding to disasters.

The role of technology is important in acquiring intelligence in a crisis situation. In the quest for intelligence for FEWRSs, a broad range of technologies such as the Internet of Things (IoT) [29], big data [30], and near-real-time satellite data [31] are used to capture critical information such as rainfall, rising river levels, and floor rates to detect flood threats. Furthermore, integrated information systems [32,33], geographic information systems (GIS) [34], and simulation techniques [35], are being used to process such information and to generate early warnings. Increasingly, crowdsource technologies based on social media [36], mobile apps [37], and Volunteer GIS [38] are being used for engaging communities in reporting incidents during the response phase. However, in order to exploit the potential of such technologies, it is important to establish a clear understanding of the "intelligence" required and the appropriate "technologies" that can be used to generate such intelligence for developing early warning and response systems.

Therefore, this paper presents the full range of intelligences needed for flood warning and response phases as well as the technologies that can be used to provide such intelligence, captured through a review of academic papers published between 2015 and 2020. The findings of this review are then synthesised to produce a conceptual architecture that illustrates how the identified intelligence and advanced technologies can be deployed to help decision makers make evidence-based decisions for early warnings and responses during flood events.

## 2. Materials and Methods

The research question established in this review is "what are the types and sources of intelligence required for effective early warning and response for flood events ?". The methodology established by Webster and Watson [39] was followed to identify and analyse the relevant literature for this review. A set of keywords was defined to search for the relevant research articles, and an inclusion/exclusion criterion was used to determine relevant and quality papers. A search criterion was established to filter relevant articles by conducting a "title" search using a combination of keywords. The keywords "floods', "response", and "warning" were used in the search since the context of this study is "floods" within the scope of the disaster management phases of "response" or "warning". Furthermore, the keywords "information" and "intelligence" were included to limit articles that are written in the specific area of interest in this review. These keywords were combined to create

the generic search string "flood" AND ("warning" OR "response") AND ("information" OR "intelligence").

The keyword combination was used in the Scopus, Web of Science, Wiley, and Gale databases, which resulted in the retrieval of 150 records. These databases allowed literature searches within a broad range of high-ranked journals and conference proceedings. Furthermore, snowball sampling was conducted, which resulted in adding 16 articles to the investigation. The overall search was limited to articles published from 2015 onwards and written in English.

Following this, the title and abstract of all papers were thoroughly examined to remove duplicates and to remove articles that are not suitable for the final analysis. This step resulted in 65 articles. In the next step, the contents of the articles were analysed to identify the relevant papers. After studying the full texts, only the articles on flood warning and response systems and processes, as well as those that describe the use of information and intelligence in the warning and response processes, were considered in this study. Papers that discussed flood hazard and risk assessments, flood preparedness, flood management, health, and other emergencies were excluded. Only articles that focused on intelligence for detecting, monitoring, and evaluating flood hazards during the warning to response stage were selected for the final analysis. This resulted in fifty-four articles (54) for the content analysis (see Figure 1). These fifty-four research articles were analysed and synthesised to extract state-of-art knowledge on the intelligence used in flood warning and response stages and the tools and techniques used to derive such intelligence.

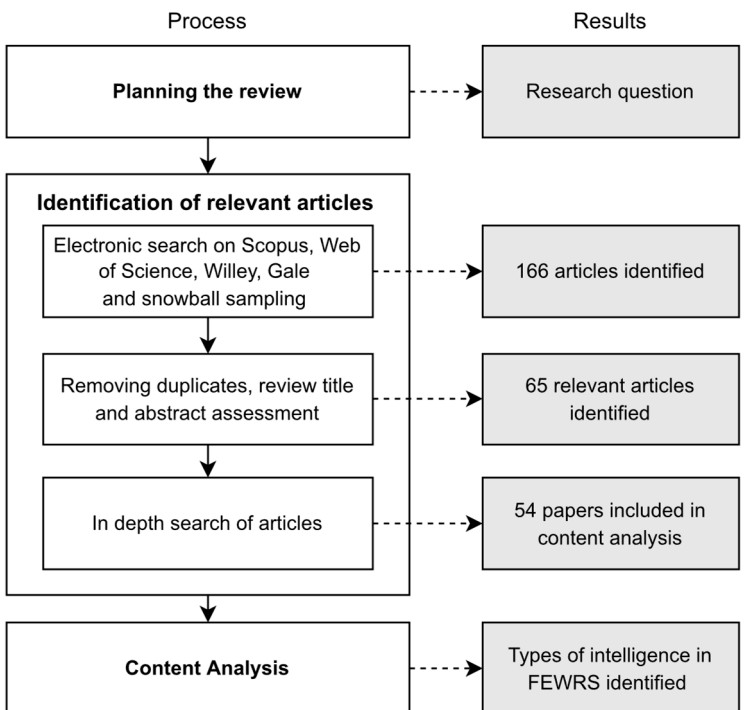

**Figure 1.** The workflow of the review process.

The scope of this review concurs with the flood risk management framework adopted by Adelekan [40]. According to Adelekan [40], planning for flood warning, evacuation, and relief are considered sub-activities in the preparedness phase, whereas emergency rescue, humanitarian assistance, and reconstruction are considered sub-activities in the response phase. Following this framework, the intelligence related to potential and historical flood inundation, damage, and losses was considered as part of the preparedness phase. Similarly, the intelligence associated with the actual flood levels, damage, and losses was considered as part of the response phase (see Figure 2).

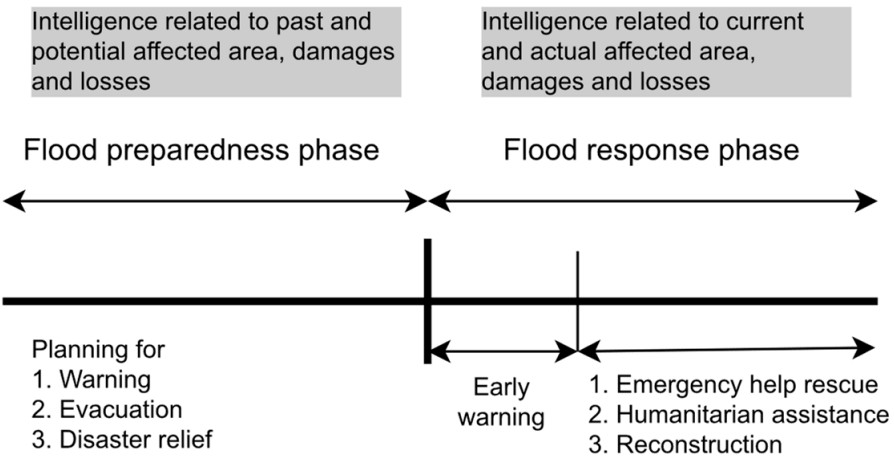

**Figure 2.** Scope of the intelligence used in the review.

Fundamental stages of early warning systems such as risk knowledge capture, monitoring and warning, dissemination and communication of warning, and preparedness to response defined by UNDRR [41] have been used to structure the review findings. The outcomes of this review are then used to establish a relationship between the flooding process, situational intelligence required at various stages, and methods (technologies) that can be used to derive intelligence to support strategic decisions at each stage of the flood warning and response process.

## 3. Results

### 3.1. Research Landscape of the Contributions

Figure 3a shows the spread of the publications used in this review between 2015 and 2020. Figure 3b shows the locations of the study, distributed across 26 countries. The main contributors were China (3 articles), the Philippines (3 articles), Pakistan (3 articles), and the USA (5 articles). However, ten contributions were either review papers or were not classified under a particular country.

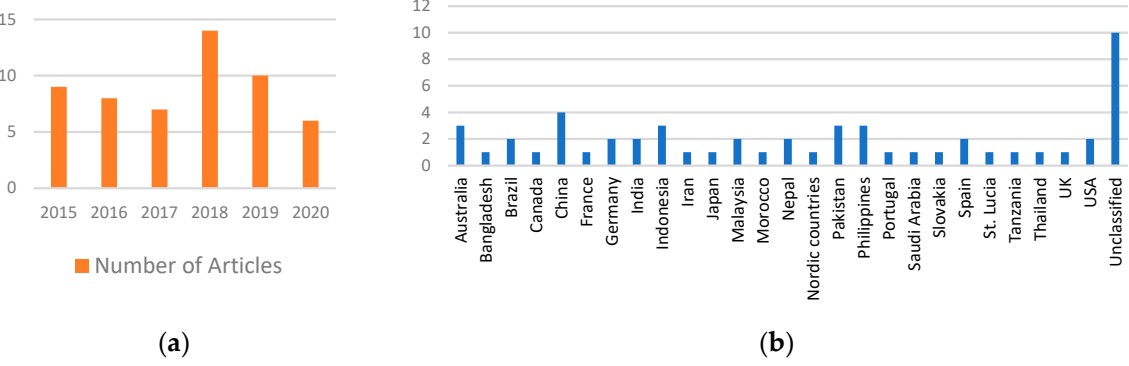

(**a**)　　　　　　　　　　　　　　　(**b**)

**Figure 3.** Distribution of articles based on (**a**) the year of publication and (**b**) the country of the case study.

### 3.2. Intelligence Used for Flood Warning and Response Phases

The types of intelligence that were identified from the review as necessary for issuing flood warnings and responses can be categorised as follows: intelligence on flood hazards; intelligence related to the population at risk; intelligence on impacted infrastructure; and intelligence on resources and capacities required during the response phase. Table 1 below summarises the types of intelligence under each category with their attributes, purpose/use, and citations.

**Table 1.** Types of intelligence used in flood warning and response processes.

| Category | Intelligence | Purpose/Use | References |
|---|---|---|---|
| Intelligence on flood hazards | Rainfall values, real-time | Flood forecasting in real time. | [36,42–48] |
| | Rainfall values, historical | Predict a possible flood scenario from past flood incidents for a given rainfall. | [49] |
| | Rainfall duration | Quantify the rainfall and forecast the floods. | [42,50] |
| | River Flow/ water flow rate (total volume passed in a given location) | Forecast floods. | [49,51] |
| | River or flood water, level measured | Assess whether the river is about to be flooded or has flooded and issue warnings accordingly. | [29,37,38,48,51–55] |
| | River or flood water, level observed | | [36,38,56] |
| | River or flood water, level forecasted | | [32,57–61] |
| | Flood inundation extent | Establish a spatial representation of floods to understand the impacted area. | [30,31,34–36,44,50,56,62–75] |
| | Flood inundation depth | Identify the hazard/risk level to the community and infrastructure. | [36,38,43,64,74] |
| | Flood intensity flood frequency/flood magnitude/ return period | Predict the hazard levels and use them to evaluate possible damage to the community, infrastructure, and natural environment. | [35,38,50,66,69,70,76–78] |
| | Historical flood events and water level | Understand inundation levels and impact caused by past flood events and extrapolate this knowledge to an emerging flood situation. | [53,56,79] |
| | Flood propagation time/lead time | Calculate lead time (travel time) of floods to plan early warnings, evacuation, and response, e.g., upstream to downstream or flood arrival time based on predicted or actual rainfall. | [44,58,59,65,69,71,72,75,76,80] |
| | Soil moisture level | Determine the level of water infiltration and flood forecasting. | [46,48] |
| Intelligence related to the population at risk | Mobility of crowd | Monitor movements of people during a disaster. | [30,62] |
| | Potentially affected population | Plan for better response, evacuation, relief distribution, and family reunification. | [30,37,62,70,77,81] |
| | Population density/demography and distribution | Useful for response planning and relief operations. | [32,68,73] |
| | Basic needs (food, water, etc.) | Acquisition and managing basic needs during the response period. | [70] |
| | Evacuation (estimated and actual) | Evacuation planning and relief management | [35,63] |
| | Affected population | Plan rescue operation and provide emergency treatments. | [37,62,63,70,82,83] |
| Intelligence related to infrastructure at risk | Potential impact on infrastructure | Develop response plan in preparedness phase to ensure efficient and effective response. | [32,64,66,84,85] |
| | Potential impact on roads | Make necessary re-routing of traffic as well as identify routes and transport methods. | [63,77,81,84] |
| | Actual impact infrastructure | Conduct actual damage assessment during and after the disaster to support ongoing response as well as future risk management and response planning. | [43,71,86,87] |
| | Actual impact roads | | [30,63,67,70] |

**Table 1.** *Cont.*

| Category | Intelligence | Purpose/Use | References |
|---|---|---|---|
| Intelligence on Resources and Capacities required during the response phase | Resources (helipads, evacuation centres, medical services, etc.) | Plan and co-coordinate response. | [73,77] |
| | Active NGOs and other voluntary organisation | Advance response planning. | [73,77] |
| | Food and Supply Information | Understand help available for humanitarian support during response. | [73] |
| | Service range (coverage) of responders (fire brigade, military, other emergency services) | Plan and coordinate response. | [81] |

### 3.3. Intelligence on Flood Hazards

The relationships between the stages of the flooding process, the numerous intelligences captured to detect and monitor each stage, and the tools and techniques that are being used to capture the intelligence are discussed in this section.

#### 3.3.1. Rainfall

Rainfall data at various point locations are typically captured through rain gauges [33,36,42–46], and the spatial variability is captured through doppler and satellite radar systems [45,47]. Such live rainfall data, as well as historical rainfall data, are used as input to the hydrological models for flood forecasting [49]. However, in cases where rain gauge data is not available, satellite observation is used to monitor and predict floods [88]. In some cases, the analyses of past flood events and their magnitudes have been used as the basis for preparing and responding to emerging flood events [45,47]. The study reported in [46] shows how the analysis of 35 years of soil moisture data derived from a satellite and integrated with gridded rainfall and elevation can be used for flood forecasting [46].

#### 3.3.2. River Water Level

The measuring of river water levels can be classified into three categories based on the method employed: measured water level (by IoT) [29,37,38,52–55], observed water level (by the public) [36,38,56], and forecasted water level through simulations [32,57–60]. Many modern early warning systems have employed IoT devices such as automated river gauges to continuously measure river water levels in real-time with greater accuracy [29,37,38,52–55]. On the other hand, active and passive social media systems and crowdsourcing platforms are also being used to report water levels observed by the community as texts and photographs with time and location data [36,38,56]. Crowdsourcing methods are beneficial for areas that do not have expensive sensor-based water level monitoring systems [38,56]. The integration of these two approaches (IoT and crowdsourcing) could increase confidence in water level measurements during disaster situations [38].

In order to gain a further lead time for issuing an early warning for an evacuation, predictive models such as hydrological models and rainfall–runoff inundation models [32,57–60] are being used to forecast water levels at a given point. The accuracy of these models can be enhanced by providing continuous real-time data gathered through both IoT and crowdsourcing [29,37,38,52–55].

#### 3.3.3. River Water Flow

River water flow is a key parameter used in hydrology that measures the amount of water passing through a specific point in time. The flow rates are typically measured by gauge stations and are used in hydrology models [49] for predicting potential floods [51].

### 3.3.4. Flood Inundation

Flood inundation extent and inundation depth are two vital types of intelligence used in flood warning and response systems. At present, near-real-time satellite data are being used to collect such intelligence during and post-event [30,31,34,44,56,62,64,71]. Radar data analysis [31] tends to be the most popular method in flood inundation mapping during the rainy season as it can penetrate clouds. In addition to satellites, airborne sensors attached to UAVs, which can supplement or even replace traditional satellite remote sensing systems, can detect the spatial coverage of floods [34].

Passive crowdsourcing media such as Twitter and Facebook [36] and active crowdsourcing platforms such as Ushahidi [63] have become popular in collecting information on flood inundation and damage [70]. People's observations of flood events in the form of photographs uploaded via social media and crowdsourcing applications have shown their value to decision making in the disaster response phase [56]. In [74], both probabilistic and deterministic approaches have been used to transform the Twitter response to floods. The review articles by Tomaszewski et al. [34] and Yu et al. [30], elaborate on how the combination of satellite and crowdsourced information can be used to determine the extent of a flood in near real time [30,34]. Two case studies from the Philippines and Pakistan reported in Jongman [71], show how the combination of multiple sources such as near-real-time satellite data and Twitter responses collected from the community, were utilised for monitoring the extent of the floods. These case studies have demonstrated how the integration of traditional remote sensing data with real-time social media data can increase situational awareness of flood hazards in terms of location, time, cause, and impact, hence improving the efficiency and speed of the response.

Many are using numerical models and GIS-based inundation mapping to determine the possible inundation zones, which allows advanced planning for disaster responses [33,35,47,49,78]. Such intelligence for response planning by disaster agencies offers sufficient time to mobilise their teams to respond efficiently and warn citizens well in advance.

Along with the inundation extent, flood depth can also be predicted before and after a flood [36,38,43,64,74] to estimate the impact on people and property by the relevant authorities [38,89]. Flood depth is typically calculated using hydrological models [38], but recently social media such as Twitter has been used to collect the flood inundation depth data [36].

### 3.3.5. Flood Arrival Time

Flood arrival time (lag time) is known as the time difference between rainfall time centroid and peak discharge [76,90]. The early prediction of the arrival time of floods at a given point is used for issuing flood early warnings to the community [44,59,65,69,71,72,75,80].

Traditionally this is measured by hydrological modelling techniques such as rainfall–runoff inundation modelling in combination with Geographic Information System (GIS) and Remote Sensing (RS) [44,65]. Recently, researchers have used intelligence from multiple sources to improve the accuracy of predicting the flood arrival time and eliminating false flood warnings. For example, Jongman et al. [71] present an approach that combines passive radar satellite data on soil moisture (AMSR) with social media data to improve accuracy in flood predictions. Similarly, Tekeli and Fouli [80] present an approach that combines AMSR satellite data with Tropical Rainfall Measuring Mission (TRMM) satellite data to improve accuracy. In [76], the authors present the analysis of historical river gauge data and the satellite data (radar) of various return periods to ascertain the lag time over a given river basin in the Charlotte Metropolitan region in the USA.

Flood arrival time is also being estimated by employing various Artificial Intelligence (AI) techniques since conventional methods are unable to capture the nonlinearity and non-stationarity related to hydrological applications [59]. Fuzzy sets and neural networks are two other popular Computational Intelligence (CI) techniques that are commonly used in the hydrology field [59]. Recent research based on the Wavelet Transform Neuro-Fuzzy (WT-NF) technique has shown promise in forecasting floods with an increased lead

time [59]. Some researchers have explored how the accuracy of the CI techniques can be enhanced by using hybrid methods that combine different CI methods for improving the accuracy and lead time of flood forecasting [59,91]. For example, [92] combines neural networks with generic algorithms, and [57] combines neural networks with the wavelet technique to increase flood forecast accuracy.

Other developments in this area include the use of Service-Oriented Architectures (SoA) [69] linked with ontological frameworks [72] for capturing and processing data from a variety of sources (IoT sensors, social media, crowdsourcing, satellites) to support the prediction of flood arrival times using the aforementioned techniques.

### 3.3.6. Flood Frequency and Return Period

Flood frequencies and return periods are two inter-related factors essential in understanding and preparing for possible situations since they indicate the magnitude of an emerging event [35,38,66,69,70,76–78]. Flood frequency analysis is a statistical technique used by hydrologists to estimate the flood return period or exceedance probability by measuring peak discharge values over a period of time. Flood frequency analyses provide decision makers with a broader understanding of the hydrological behaviours of a given river from the perspective of the flood response [76]. Higher peak discharge and runoff rates increase flood frequency, which increases the severity of floods. Therefore, it is necessary to understand the flood hazard level at different flow conditions so that proper evacuation planning can be arranged in advance [35]. In addition to the frequency calculation, historical flood events are useful for validating various models, developing risk and damage assessment techniques, and preparing for future events [53,56,79].

### 3.4. Intelligence Related to Exposed Population

The intelligence required to understand and estimate the exposed population and the underpinning technology that can be used to acquire such intelligence during flood hazards are discussed in this section.

### 3.4.1. Population Density, Distribution, and Demography

The spatial distribution and density of populations are primary data that are required in order to identify and estimate an exposed population for a given hazard [32,68,73,93]. Population data are usually obtained from the national census, available in spatially aggregated forms up to local administrative boundaries, which are too coarse for disaster impact analyses. Hence, land use maps [93] and satellite-derived settlement data [94] are being used to derive population density maps at finer scales. In addition, global data sources such as the Landscan data also provide population grids of various grid sizes [95].

### 3.4.2. Potentially Affected Population

The potentially affected population by a flood is the most important intelligence required by authorities to make decisions during the early warning and response stages [30,37,62,70,77,81]. Furthermore, an estimation of the affected population is essential to plan for relief assistance and post-disaster impact assessments [70,77]. Data from various sources such as government authorities and municipalities are typically combined with open-source spatial data to estimate the exposed population in the GIS domain [81]. Tzavella et al. [81] report how Volunteered Geographic Information (VGI) has successfully been used in an extreme flood event in Cologne, Germany, to improve the efficiency of the flood response with decreased response times.

Numerous models and approaches have been used to evaluate the potential effects of floods on people. For example, the Disaster Diagnostic and Evaluation System (SEDD) offers a fuzzy rule-based classification system that can be used to assess the possible impact on people just after a disaster [77]. It uses the Emergency Events Database (EM-DAT) as the primary source of population data together with sources such as the Human Development Index (HDI), published by UNDP, to calculate vulnerabilities. Deng et al. [70] propose a

social media-based model to estimate the impact of a disaster on the community, which has been tested for typhoon Haiyan. In contrast, Ushahidi collects actual affected population data during the Haiti earthquake [62] using crowdsourcing.

### 3.4.3. Mobility of Crowd

Intelligence with respect to the location and mobility of crowds is critically important in the emergency response phase and allows authorities to target people who need immediate rescue and medical assistance. Call Detail Records (CDR), referred to as digital trails of modern mobile device users, can be used to monitor population movement and displacement and in disaster response planning [62] since it offers a detailed record of mobile phone locations and call logs generated by mobile companies in real time. The successful use of CDR techniques is reported in [30] during the Haiti earthquake. Even though CDR is a useful technology for understanding population dynamics, it is still not widely used due to privacy issues and lack of supportive legal frameworks [62].

### 3.4.4. Evacuation Needs (Estimated and Actual)

People who require evacuation or have already been evacuated are another critical type of intelligence useful in the response phase. The number of people who require evacuation is typically estimated and identified during the preparedness planning process for various flood simulation scenarios for multiple return periods [35]. However, a more accurate picture of evacuated people can be captured through social media platforms, active and passive crowdsourcing, and geo-referenced VGI techniques during a disaster [62].

### 3.4.5. Affected Population

Intelligence on affected people such as those who are trapped, injured, and victims who need immediate rescue is critical during an emergency response. Furthermore, they require a mechanism to connect with response teams and their families and friends who are concerned about their safety and well-being.

Crowdsourcing applications [62], social media microblogs [70,83], and mobile CDR [62] are potential tools and technologies used to gather the status and needs of affected people in real time. As successfully demonstrated during Typhoon Haiyan, a semantic analysis of the microblog posted on social media can help authorities to understand the concerns of affected people at different stages of the disaster and respond better [50]. Ushahidi is another popular crowdsource application that has been successfully used to collect, visualise, and map data gathered from affected communities [62].

Eivazy and Malek [82] illustrate an example of how agent-based solutions, integrated with crowdsourcing services, were used during the Aquala flood disaster in Iran in 2019 to help victims obtain emergency support from rescuers. In this example, individuals injured and in a critical situation are reported through crowdsourcing systems, and an agent-based information system attempts to ensure the victim's safety by connecting them with rescuers [82]. The increasing trend in providing safety checks through social media systems such as Facebook to inform friends and family during a disaster is now common and reported in [62]. Bachmann et al. [37] present a mobile app that can be used to reunify families affected by disasters.

### 3.4.6. Essential Needs

During the response phase, government authorities are also responsible for supplying essential needs such as the food and water required by displaced populations. Intelligence regarding essential needs is typically collected from microblogs such as Twitter [70,83], social media, and crowdsourcing systems [30]. Deng et al. [70] report that during Typhoon Haiyan, a community in Hainan, China, used social media techniques ("Sina Weibo", a Chinese microblog similar to Twitter) and semantic analysis to inform the relevant authorities of the needs of the affected people.

*3.5. Intelligence Related to Affected Infrastructure*

3.5.1. Potential Impact on Infrastructure

The potential impact of floods on infrastructure, buildings [64,66,84,85], and roads [63,77,81,84] is essential intelligence required for disaster preparedness and response. Geo-referenced data on buildings, critical infrastructure, and road networks obtained from administrative sources and using VGI techniques, including OpenStreetMap, are integrated with flood inundations maps and can be used to obtain information on infrastructure exposed to the floods [32,84].

Potential damage to residential buildings and other infrastructure is typically estimated using simulation techniques for multiple return periods with different exceedance probabilities of floods [64,66,85]. Vulnerability curves that represent the damage levels to buildings for different levels of floods are used to assess possible damage to buildings and to propose hard and soft mitigation solutions [66]. The monetary value of the damage is then aggregated at different scales, from an individual building to administrative boundaries and catchment areas [85]. In addition, the early identification of road inundation possibilities allows authorities to explore different re-routing options during a disaster [81].

3.5.2. Affected Infrastructure

Intelligence regarding the actual impact on infrastructure, both during and after a disaster situation, is essential in managing disaster situations. The use of near-real-time satellite data and social media responses (Tweets) for calculating such intelligence is reported in [71]. Similarly, the use of geo-tagged images of damaged buildings to conduct damage assessment is reported by Bica et al. [86] and Nguyen et al. [87]. Based on a study conducted in Nepal by Bica et al. [86], a positive correlation has been observed between actual ground damage and damage assessment results conducted using geo-tagged Twitter responses during earthquakes that occurred in April and May 2015.

The analysis of historical damage data from multiple flood events provides a comprehensive view of past flood damage. In [43], the authors present a comprehensive database that captures actual damage to housing, infrastructure, and the economy for various historical flood events that can be used for future mitigation and response planning processes.

Intelligence regarding inundated road networks is necessary during the emergency response phase to plan and re-route rescue services as well as establish regular transportation. Road inundation data during floods is acquired mainly by social media, crowdsourcing, near-real-time satellites, and UAV [30,63,67,70].

*3.6. Intelligence on Resources and Capacities*

Resources and Capacities

Intelligence on available resources and capacities are required in order to respond to disasters [73,77], including available response organisations and volunteers [73], health services [96], and food and supply information [73]. Saad et al. [73] present the successful implementation of an Integrated Flood Disaster Management system in the District of Kemaman, Malaysia, that is comprised of a database containing the critical resources and capacities required during a flood response. In their system, intelligence, such as details of evacuation centres, data on non-Governmental Organisations (NGO) and other volunteer organisations, and data on helipad locations have been identified as necessary capacities during responses in order to manage logistics to transport food and essential needs as well as efficient response management [73]. The locations of these facilities are typically organised and stored in GIS databases.

The locations of health facilities and the travel times to such facilities are considered useful intelligence in the emergency response phase to manage flood-affected victims [96]. OpenStreetMap (OSM)-derived global health facility data with their locations and other attributes are made available via www.healthsites.io. In [97], access to healthcare facilities has been analysed and presented in global maps to visualise travel times by foot and motorised transport.

Tzavella et al. [81] calculate the service range of the first responders such as the fire brigade, through network analysis, taking into account the road network, points of resources, and floods in Cologne, Germany.

## 4. Discussion

The critical analysis of the literature shows that the situational intelligence obtained for flood warning and responses is associated with rainfall, river flow, inundation, impact on people, properties, and response capacities. It was observed that numerous tools and technologies are used to derive intelligence that is transformed into decisions. The relationship between the flooding process, the required intelligence, and the tools and technology to derive this intelligence can be presented as a conceptual system architecture for making informed decisions for early warnings and responses. This conceptual architecture can be presented in four key segments for ease of understanding, as discussed below.

### 4.1. Conceptual Model of Flooding Process and Warning Generation

According to the literature, it was observed that numerous technological approaches such as IoT [29,37,38,52–55], crowdsourcing [36,38,74], satellites [31,44,64,71,80,98], and numerical modelling [35,58,59,85] are used to extract intelligence related to flooding at various stages, such as rainfall, river flow propagation, and inundation as indicated in Figure 4.

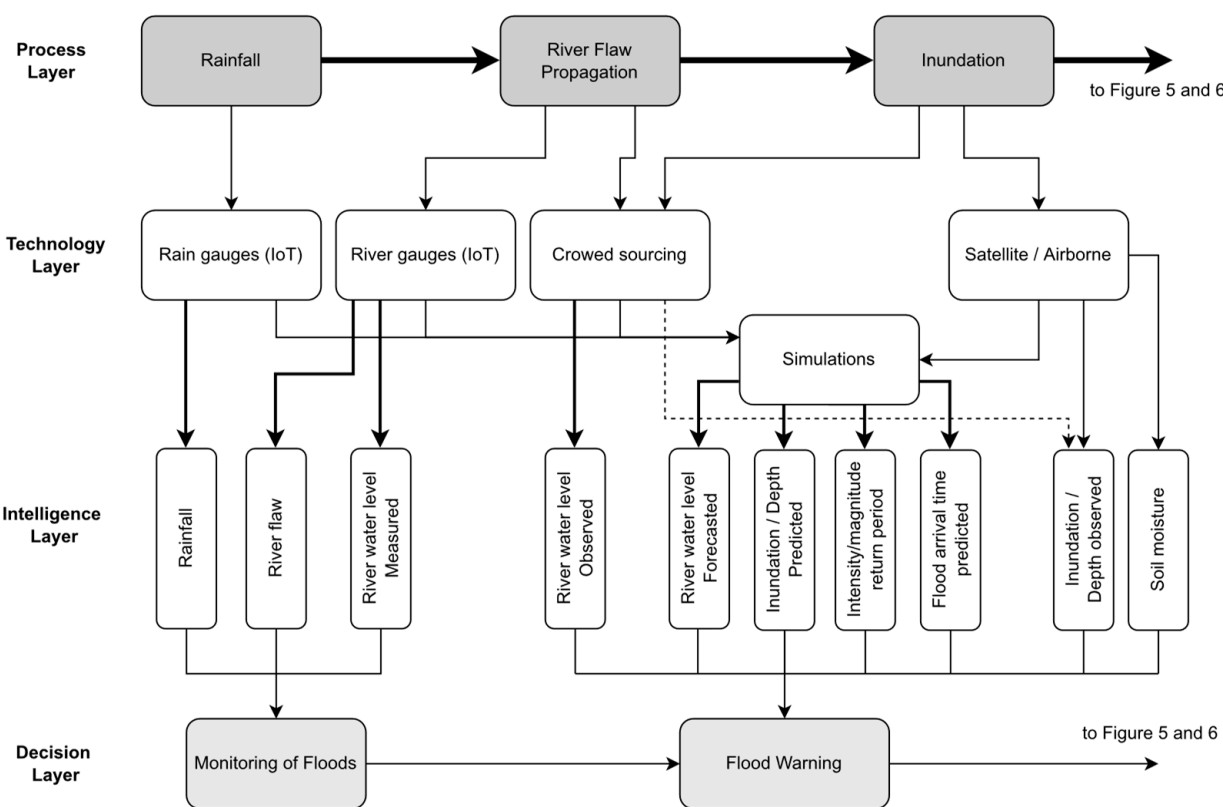

**Figure 4.** Intelligence required for monitoring emerging flood situations.

The intelligence extracted from these technologies includes rainfall, river level (measured, observed, and forecasted), both inundation depth and extent (measured, observed, and forecasted), flood frequency, return period, intensity, flood arrival time, and soil moisture. Figure 4 captures the use of technological approaches for extracting intelligence to respond to various activities by disaster management personnel during a flood disaster scenario. The overall conceptual architecture presented in Figure 4 integrates four layers: the process layer, technology layer, intelligence layer, and decision layer. The process layer represents how the flooding process evolves starting from the rainfall and river flow

to inundation. The technology layer can then be built using the technological solutions identified in this survey to monitor the evolving flooding situation and extract and pass the relevant information to the intelligence layer. The information captured in the intelligence layer can then be used by disaster management authorities to monitor the evolving flood situation and generate flood early warnings in advance, as illustrated in the decision layer. The conceptual architecture presented in Figure 4 can be implemented using the state-of-the-art technology presented in the previous sections to extract the relevant intelligence, allowing decision makers to ensure public safety before, during, and after floods.

However, it should be noted that there are many barriers to implementing such systems [16,99,100]. Some barriers and challenges include (i) the inadequate coverage of IoT sensors due to capital and maintenance costs and the unavailability of internet connections [16], (ii) the lack of accurate flood simulation models running on high-performance computers to provide near-real-time responses [99], and (iii) the limitations of acquisition and the limited coverage of near-real-time satellite images [100]. Although many developing countries have access to the International Charter for Space and Major Disasters, Copernicus System, and Sentinel Asia System, the average time for satellite activation for receiving the first image is three to four days [101]. As a result, many disaster management agencies in developing countries resort to historical inundation information to estimate the possible inundation zones during flooding incidents. In this context, crowdsourcing techniques are more efficient than satellite observations, even with the limitations of their effectiveness and accuracy [102].

### 4.2. Conceptual Model of Flooding Impact on People

When a population is exposed to floods, intelligence such as the movements of people and their vulnerabilities and numbers, as well as the locations of people trapped or injured, people evacuated, and their basic needs are required by authorities and response teams. These are acquired during different phases of the disaster event (before, during, and after) using simulations, crowdsourcing techniques, voluntary GIS activities, social media, call detail records (CDR), and remote sensing.

Figure 5 illustrates the relationship between the impact of flood inundation on people and the technologies that can be used to derive intelligence for supporting evacuation and rescue operations. As shown in Figure 5, as the inundation is impacting the population, people will begin to self-evacuate, sometimes with the support of government agencies and NGOs who need to evacuate vulnerable people with mobility and health conditions. Following the same layered approach used in Figure 4, Figure 5 shows how the various technology solutions identified in this survey can be used to extract the intelligence required for issuing early warnings and conducting intelligence-driven rescue and relief operations as the inundation is impacting people, as shown in the process layer.

The flood inundation results derived from simulations and satellites, and overlayed with census data have the potential to provide intelligence on potentially affected people and those who are at risk. Such information can be used to disseminate targeted warnings to the people at risk before the floods, hence saving lives. As the flood begins to impact people, technologies such as CDR, crowdsourcing, and social media techniques can be utilised to gain near-real-time intelligence on affected people on the ground to coordinate evacuation and rescue operations.

However, access to up-to-date population data is problematic since the population distribution and demography are obtained mainly from the national census, where most countries typically release such data sets in 10-year intervals. As a result, the population growth in the in-between years is not captured by these censuses. Furthermore, the national census registers do not usually capture the population dynamics at workplaces, schools, hospitals, hospices, and other public localities. Hence, census data alone will not provide the actual situation on the ground to estimate potentially affected populations during a flooding situation. Hence, there is a need for local representatives to maintain a more comprehensive database of their local populations in order to better respond to disasters.

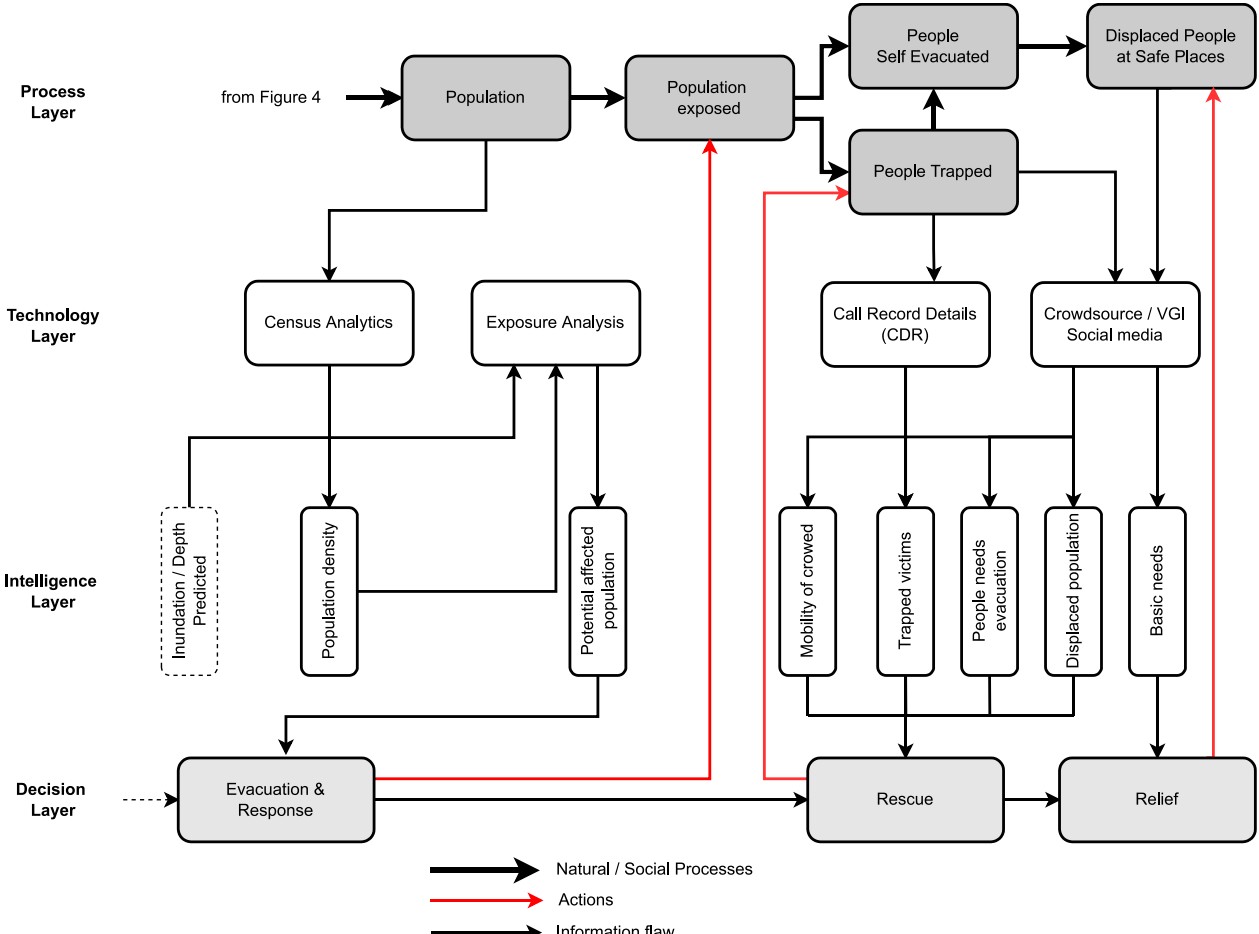

**Figure 5.** Intelligence required for issuing early warnings, rescue, and relief operations.

On the other hand, the accuracy of the predicted inundation scenario plays a vital role in determining the affected population. Therefore, simulation models used during disaster situations should be calibrated and validated well in advance to ensure the accuracy of their outputs.

Even though social media and crowdsourcing techniques exist, these systems are not standardised and well-recognised in disaster response plans at a local level [103]. Furthermore, at present, community participation is not actively encouraged to realise the maximum benefit of these techniques. Although CDR technology has the potential to offer active mobile SIM card locations and the movement of people at risk during a disaster [62], such information is typically not available due to privacy issues. The exploitation of these possibilities would require disaster management agencies to work closely with the mobile service providers and integrate them with their current disaster response processes while providing a legal framework for accessing such private data for emergency purposes.

### 4.3. Conceptual Model of Flood Impact on Infrastructure

Intelligence on physical properties such as housing, utilities, other infrastructure, and road networks that could be affected by floods is required by authorities for optimum risk management planning and response. These intelligence needs can be classified into two categories: (i) pre-disaster intelligence on infrastructure that could potentially be affected, and (ii) intelligence on the affected infrastructure both during a disaster and in the post-disaster phase.

Figure 6 presents a layered approach that represents the relationship between the impact of flood inundation on infrastructure and the potential technology that can be used to derive intelligence to support decisions. As in the previous sections, the layered

architecture is represented through the process layer, technology layer, intelligence layer, and decision layer. The infrastructure that can potentially be impacted by floods is usually identified through exposure analysis using the infrastructure data collected from various government agencies together with the estimated inundation. This intelligence can be used for advanced evacuation planning, safeguarding household items and livestock, and building mitigation plans and business continuation plans for infrastructure (utility, public services, government buildings, and economic centres).

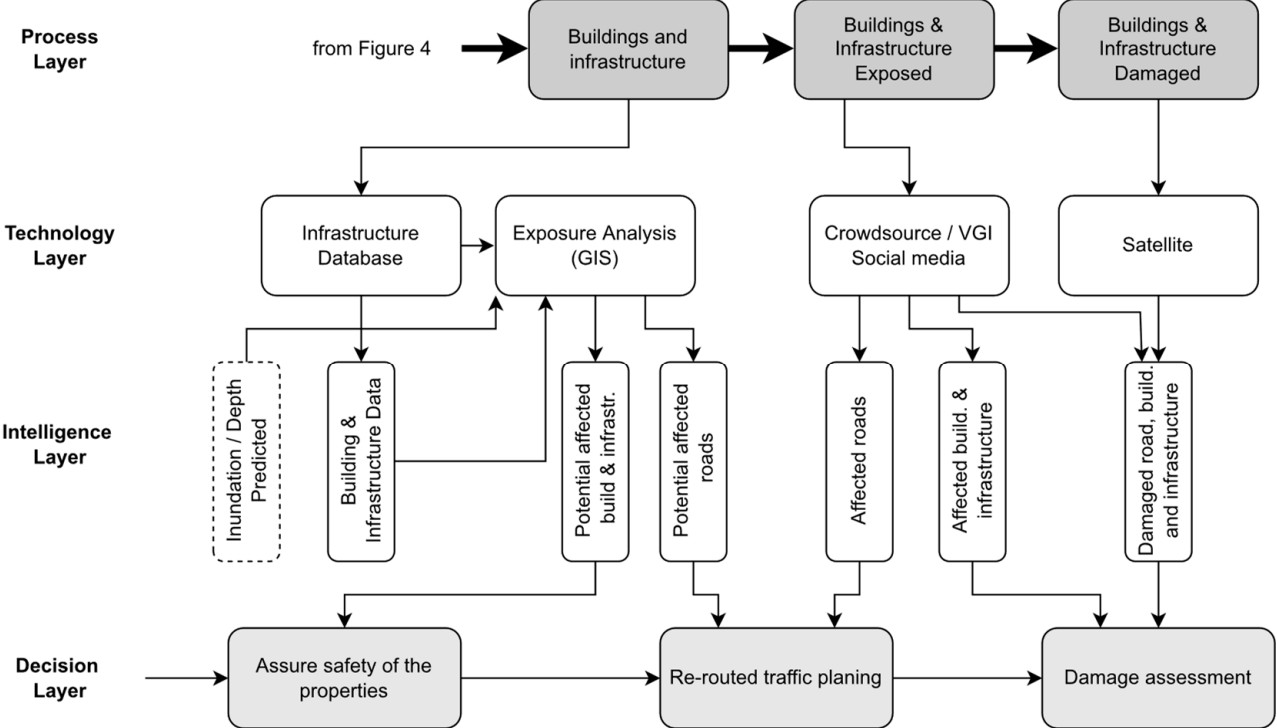

**Figure 6.** Intelligence required for identifying affected infrastructure.

Although the above flood preparedness plans allow authorities to identify the potential risks to infrastructure and implement mitigation measures using existing data, sources such as social media, crowdsourcing technologies, and satellite imagery are important to establish the actual situation on the ground during a disaster. However, the use of satellite images for the response is still challenging as the acquisition and derivation of intelligence from such sources requires considerable time [104].

*4.4. Conceptual Model of Response Capabilities*

Intelligence on the resources and capacities required for a successful response is necessary for the authorities to make timely coordination with the relevant parties. For example, situational intelligence on safe centre locations and their capacities, evacuation routes, transport facilities, as well as the locations of affected people are essential for effective evacuation planning. Novel process models could optimise mass-scale evacuation planning by efficiently coordinating resources with affected communities [105]. Furthermore, authorities also require information on surge capacities for food, medical assistance, transportation, and the availability of volunteers for the better coordination of evacuations. Hospital evacuations need extra attention as patients are one of the most vulnerable groups during flood emergencies. State-of-the-art hospital evacuation models can be used as potential solutions to safeguard patients' safety during floods [106].

Figure 7 illustrates the layered approach where intelligence on capacities and resources can be obtained through numerous resource management databases and systems to assist in the decision-making process. More specifically, during a flood emergency, authorities need

to locate the nearest evacuation centres and health facilities with the appropriate capacities that match the requirements to re-locate displaced or treat injured persons. Typically, local flood preparedness plans identify such facilities and hosting capacities well in advance. In addition, volunteers, volunteer agencies, and other resources such as transport, heavy machines, and tools are required to respond on demand.

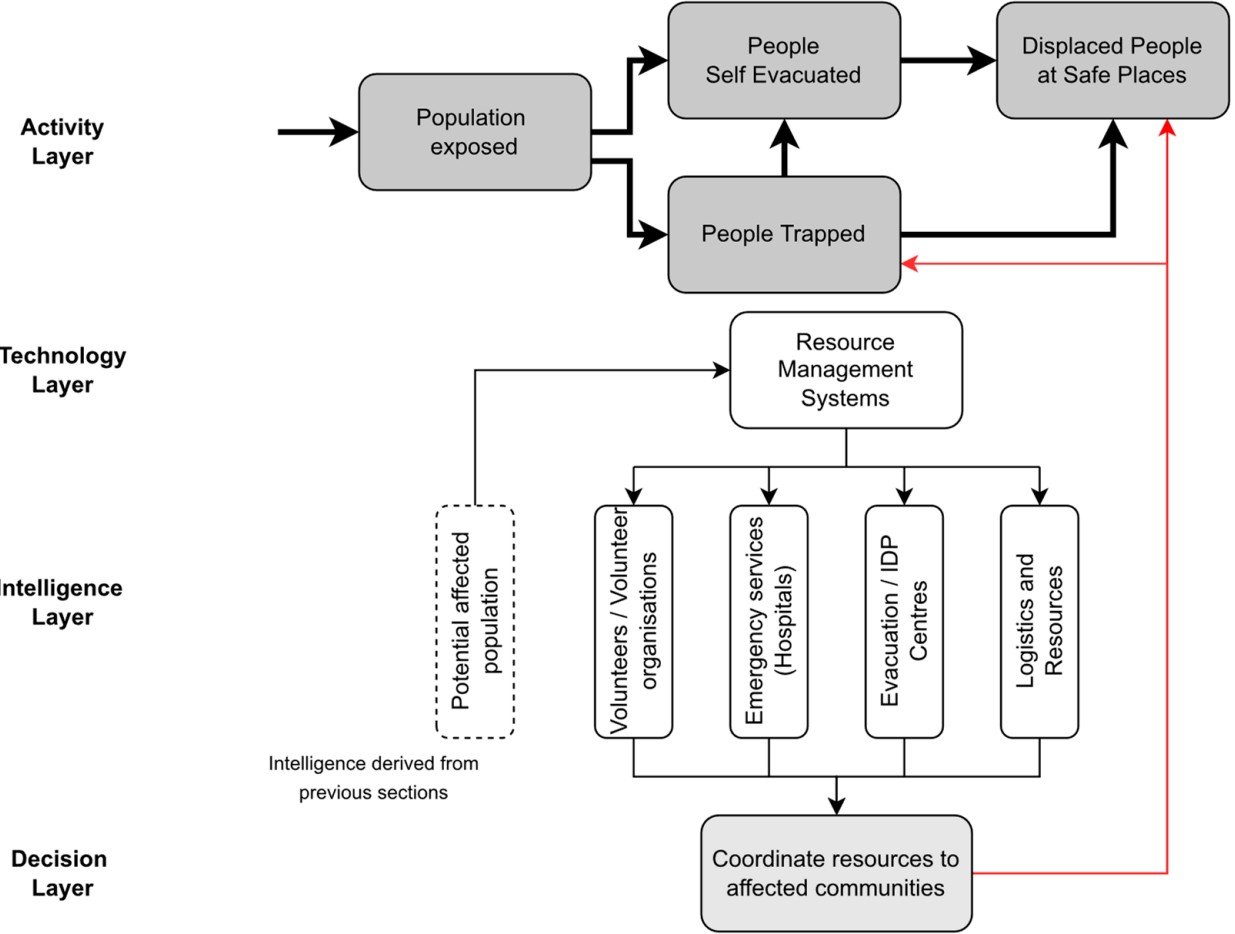

**Figure 7.** Intelligence required for capacities and resources in responses.

## 5. Conclusions

The review of the literature presented in this paper identified twenty-eight types of intelligence necessary during various stages of the FEWRS (pre-flood, during the flood, and post-flood) to issue flood warnings in advance and to respond efficiently to safeguard people and property. Over 54 published articles from several bodies of knowledge, including information systems, disaster risk management, and hydrometeorology, have been examined to establish a relationship between the flooding phenomena, the intelligence required for evidence-based decision making, and the sources of technology that can be used to extract such intelligence.

The precondition for extracting critical intelligence during a flood situation is the availability of exposure and vulnerability data of people and infrastructure in the flood-prone area under consideration. As the flood situation begins to develop, real-time information regarding the flood hazard can be captured using numerous techniques and tools: citizens as sensors, satellite remote sensing technology, IoT devices, and mobile devices. Information from citizens can be captured through social media and crowdsourcing techniques. These raw data can then be used by GIS, artificial intelligence (AI), or hydro-dynamic modelling to extract critical intelligence such as the dynamic characteristics of the hazard (rainfall,

river water level/flaw, flood arrival time), the population and infrastructure exposed or at risk, and the capacities required during a response as presented in Table 1.

The conceptual architecture presented in this paper provided guidance for deploying various advanced technology approaches for deriving the necessary intelligence required by disaster management agencies as the floods begin to spread and impact the community and the environment. The architecture presented in Figures 4–7 illustrated how the required intelligence during the flood cycle needs to be managed in order to inform, evacuate, rescue, and offer relief to citizens and promptly safeguard property.

Moving forward, the layered approach presented in this paper offers a foundation for developing a technology platform that disaster management agencies can use to issue early warnings with sufficient time for people to evacuate, better respond during floods, and efficiently manage relief operations. Furthermore, the conceptual system architecture presents a range of technology solutions that can be adopted by decision makers based on the availability of the technology and offers a pathway to increase the accuracy and efficiency of receiving the necessary intelligence as the resources become available. It shows how information from sensors, databases, big data systems, GIS, hydrological simulations, and satellite remote sensing can be combined to offer a rich set of data for decision making and interventions by various agencies. The integration of these technologies has the potential to increase the effectiveness, efficiencies, and accuracy of the overall approach to flood monitoring and early warning, and evacuation.

The proposed integration will overcome the limitations of the present early warning and response systems such as the unavailability of information and intelligence [9]; insufficient information sharing [107–110]; lack of coordination among agencies [111,112]; false early warnings [113]; lack of allocation of resources for responses [111]; and delayed responses [114], which often result in a crisis escalation and a higher numbers of causalities.

The authors acknowledge that the keyword combinations deployed in this research for searching academic papers may have missed some important articles. As a result, some important intelligence required for early warning and response systems, and novel technologies that can offer such intelligence may have been omitted in this study.

**Author Contributions:** Conceptualisation, methodology, formal analysis, and writing: S.S.; review and editing, T.F. and B.I.; project administration and funding acquisition, T.F. All authors have read and agreed to the published version of the manuscript.

**Funding:** This research was funded by Global Challenges Research Fund (GCRF) and the Engineering and Physical Sciences Research Council (EPSRC), through the project "A Collaborative Multi-Agency Platform for Building Resilient Communities", grant number EP/PO28543/1.

**Institutional Review Board Statement:** Not applicable.

**Informed Consent Statement:** Not applicable.

**Data Availability Statement:** Not applicable.

**Acknowledgments:** The authors express their gratitude to the Global Challenges Research Fund (GCRF) and the Engineering and Physical Sciences Research Council (EPSRC) for the financial support under the International Grant, EP/PO28543/1, entitled "A Collaborative Multi-Agency Platform for Building Resilient Communities".

**Conflicts of Interest:** The authors declare no conflict of interest.

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
