# Peer review of "Advanced Technologies for Offering Situational Intelligence in Flood Warning and Response Systems: A Literature Review"

_water, doi:10.3390/w14132091_

Round 1

Reviewer 1 Report

This paper solved the question of what are the types and sources of intelligence required for effective early warning and response for flood events. However, the writing of this article is not standardized, there are a large number of typing errors and title errors, and the definition of ‘intelligence’ discussed is vague and lack of literature support. In addition, there are the following specific shortcomings.

(1) As a review article, the steps of literature screening are chaotic. The first paragraph on page 3 suggests re-editing after clarifying the idea.

(2) The results in Table 1 are expressed incorrectly. ‘06’ should be ‘6’.

(3) Note the typo in the article. There is an extra ‘)’ in ‘preparedness to response defined by UNDRR [27]) are used to structure the review findings.’ in Line 1 of Page 4. There is an extra ‘)’ in ‘Attributes)’ of Table 2.

(4) Whether the first and second columns in Table 2 are wrong. They are both ‘intelligence on flood hazards’. In addition, pay attention to format consistency. Whether the contents in the table should be bold.

(5) The titles of Section 4 and Section 5 are the same.

(6) There are two subsections named Section 3.4.

(7) Whether a single word or phrase as a line meets the specification. Please adjust the corresponding parts of sections 3.3 to 3.5.

(8) Figures 1, 5 and 8 are very blurred.

(9) What is the definition of ‘intelligence’? What is the difference between it and ‘technology’? There is not much introduction to ‘intelligence’ in the literature, and its definition is very vague.

(10) The reference is malformed. The ‘year of publication’ should be in bold, not the ‘volume’.

(11) The titles of the figures do not have to be all capitalized.

(12) The contribution of this paper is insufficient. This paper just classifies the existing literature according to ‘intelligence’. The definition of ‘intelligence’ is obscure, and the relationship between several categories is not clear.

Author Response

File is uploaded

Reviewer 2 Report

Please see attached file for details

Author Response

A file is attached

Reviewer 3 Report

The title of the paper should be polished. Particularly “Intelligence Required” is a bit awkward.

Please revise this claim in the abstract “ Losses from floods can be reduced by having accurate intelligence of an emerging flood situation in order to make timely decisions for issuing early warnings and responding efficiently.”

Many terms like “ key intelligence required” have been used in the paper. These terms are a bit obscure.

The keywords “intelligence” and “information” are very generic. Please use more specific keywords.

Please avoid using acronyms in the keywords.

Several initial sentences of the introduction should be supported by appropriate references.

I can see in some sections that “natural disaster(s)” has been used. In contrast, this term is not acceptable in disaster science anymore. Instead of that, currently, “natural hazard” is used. Please revise the whole paper based on this comment.

Authors claim,”  In this regard, Sendai Framework for Disaster Risk Reduction (SFDRR) emphasises the need for the availability of multi-hazard warning systems and disaster risk information to the community by the end of 2030.” Please mention the page and the line of this claim in the Sendai framework.

Page 2, “Intelligence is crucial for making sound decisions.” Please revise this claim. It is not clear.

The authors should clearly discuss the research contributions at the end of the introduction.

Some studies have been missed in the introduction. At least the authors can discuss similar studies focused on developing flood warning systems and their structures and basic requirements.

The reference style should be consistent. I can see, for example, in section 2, the authors used “Webster and Watson (2002)”

The research method is a bit under question. The authors mentioned that they used “Information” and “Intelligence” to limit the number of studies. While this procedure may exclude many studies.

The authors stated “The keyword combination was used on Scopus, Web of Science, Wiley, Springer, Science Direct and Gale databases” this is not very professional. For example, almost all papers which are in ScienceDirect are covered by Scopus, except those journals that are very new and have been established in less than two years.

For finding papers, it is recommended that the authors use a flowchart like the flowchart that has been provided in: "Hospital evacuation modelling: A critical literature review on current knowledge and research gaps." International Journal of Disaster Risk Reduction 66 (2021): 102627.

Please revise table 1. It seems not correct. The google scholar has found 16, while the Scopus has found 47? Almost all papers that are in Scopus can be found on google scholar.

The quality of figure 1 is very low; please provide a high-quality figure.

Figure 2 is very obscure. It is not clear what was the aim of the authors. I can not see any specific logicbehindf this figure. There are many “intelligence” boxes.

How the authors reached to the figure 2. It seems this figure has been formed based on the knowledge of the authors instead f extracting from a systemic process.

The section 3 starts with “This section may be divided by subheadings. It should provide a concise and precise description of the experimental results, their interpretation, as well as the experimental conclusions that can be drawn.” Is it a part of the paper? It is more like a comment that the authors forgot to delete!

Figures 3 and 4 can be presented side by side. Allocating about 1 page to these figures is not acceptable.

In subsections 3.3, please use a numbering system for “rainfall” , river water level” , …

Similar to the above comment for section 3.4

Please discuss the possible implications of this study, partuclary in evacuation planning. Please support your discussion by using the following references:

"Dueling emergencies: flood evacuation ridesharing during the COVID-19 pandemic." Transportation research interdisciplinary perspectives 10 (2021): 100352.

"Enhancing evacuation response to extreme weather disasters using public transportation systems: a novel simheuristic approach." Journal of Computational Design and Engineering 7.2 (2020): 195-210.

"Enhancing pedestrian evacuation routes during flood events." Natural Hazards (2022): 1-25.

"A modelling framework to design an evacuation support system for healthcare infrastructures in response to major flood events." Progress in Disaster Science 13 (2022): 100218.

The resolution of figure 5 is very low.

The title of section 4 should be changed. It is very similar to section 5.

Author Response

A file is attached

Round 2

Reviewer 1 Report

The authors addressed all the comments. I accept the paper in its present form.

Reviewer 3 Report

The paper has been improved significantly.